# Predicting the spatio-temporal spread of West Nile virus in Europe

**José-María García-Carrasco**[1]*, **Antonio-Román Muñoz**[1], **Jesús Olivero**[1], **Marina Segura**[2], **Raimundo Real**[1]

**1** Biogeography, Diversity and Conservation Lab, Department of Animal Biology, Faculty of Sciences, University of Málaga, Málaga, Spain, **2** International Vaccination Center of Malaga, Maritime Port of Malaga, Ministry of Health, Consumption and Social Welfare, Government of Spain, Málaga, Spain

* jmgc@uma.es

**Data Availability Statement:** All relevant data are within the manuscript and its Supporting Information files.

**Funding:** J.M.G.C. received a FPU grant from the Ministerio de Educación, Cultura y Deporte

## Abstract

West Nile virus is a widely spread arthropod-born virus, which has mosquitoes as vectors and birds as reservoirs. Humans, as dead-end hosts of the virus, may suffer West Nile Fever (WNF), which sometimes leads to death. In Europe, the first large-scale epidemic of WNF occurred in 1996 in Romania. Since then, human cases have increased in the continent, where the highest number of cases occurred in 2018. Using the location of WNF cases in 2017 and favorability models, we developed two risk models, one environmental and the other spatio-environmental, and tested their capacity to predict in 2018: 1) the location of WNF; 2) the intensity of the outbreaks (i.e. the number of confirmed human cases); and 3) the imminence of the cases (i.e. the Julian week in which the first case occurred). We found that climatic variables (the maximum temperature of the warmest month and the annual temperature range), human-related variables (rain-fed agriculture, the density of poultry and horses), and topo-hydrographic variables (the presence of rivers and altitude) were the best environmental predictors of WNF outbreaks in Europe. The spatio-environmental model was the most useful in predicting the location of WNF outbreaks, which suggests that a spatial structure, probably related to bird migration routes, has a role in the geographical pattern of WNF in Europe. Both the intensity of cases and their imminence were best predicted using the environmental model, suggesting that these features of the disease are linked to the environmental characteristics of the areas. We highlight the relevance of river basins in the propagation dynamics of the disease, as outbreaks started in the lower parts of the river basins, from where WNF spread towards the upper parts. Therefore, river basins should be considered as operational geographic units for the public health management of the disease.

## Author summary

West Nile virus is a widely spread flavivirus, which is transmitted from birds to humans by mosquitoes. In humans, the virus can cause West Nile Fever (WNF) and in some cases, it affects the nervous system leading to severe symptoms that may result in death. Human

(FPU17/02834) (http://www.educacionyfp.gob.es/portada.html). J.O. is supported by the Project CGL2016-76747-R from the Ministerio de Economía, Industria y Competitividad and FEDER Funds (https://www.mineco.gob.es/portal/site/mineco/). A.-R.M. is supported by project UMA18-FEDERJA-276 (Programa Operativo FEDER, Consejería de Economía, Conocimiento, Empresas y Universidad, Junta de Andalucía). The funders had no role in study design, data collection and analysis, decision to publish, or preparation of the manuscript.

**Competing interests:** The authors have declared that no competing interests exist.

cases have increased in Europe since the large-scale epidemic in 1996, being 2018 the year with the highest number of cases registered to date. We developed risk models based on 2017 cases, and predicted in 2018: 1) the occurrence of the disease; 2) the intensity of the outbreaks; and 3) the imminence of the cases. We identified favorable areas for the incidence of the virus in which environmental and human-related variables had an important role. The outbreaks began in the lower areas of large river basins and spread to higher areas, which highlights the importance of river basins in the propagation of outbreaks. Consequently, the early warning should be based on a basin scale.

## Introduction

West Nile virus (WNV) is an emerging *Flavivirus* (family *Flaviviridae*) [1] transmitted by different mosquito species of the genus *Culex* and *Aedes* [2,3]. The virus is maintained in an enzootic cycle among birds, which are the WNV competent reservoirs [4]. Humans and equids are the dead-end hosts of the virus cycle [5]. When infected by WNV, most humans remain asymptomatic, whereas 20% develop a zoonotic febrile illness [5] that is known as West Nile Fever (WNF). Less than 1% of infected individuals develop severe neurological symptoms, with a 0,1% mortality rate of those infected [6]. Today there is still no human vaccine or antiviral treatments for WNV, which makes identifying the environmental correlates of the disease more important for prevention [7].

WNV is currently spreading in Europe [8]. From its isolation in 1937 in the West Nile District of Uganda [9], it is thought to have spread throughout the globe via migratory birds [8]. It is also supposed to have passed in this way from Africa to Europe, where it is circulating since the 1950s, with the first European outbreak in humans recorded in 1962 in the Camargue, southern France [10]. In 1996, Europe experienced its first major WNV infection epidemic, with 393 cases in Romania, including 17 deaths [11]. Since then, new cases have been reported in eastern, western and southern Europe, with a typical seasonal nature from April to November [5]. The transmission season of 2018 was exceptional; 1,605 cases were confirmed, which is double the sum of cases registered in the previous three years [12,13]. Although WNV has been detected widely in birds and horses through Europe [14], the distribution of the disease in humans is restricted to the southern and southeastern Europe.

Understanding the processes that drive the distribution patterns of living organisms in space and time is a main aim of biogeography, which, thus, may provide useful insights to medical geography [15]. The study of spatio-temporal patterns in the distribution of diseases has recently been termed as pathogeography [15]. Zoonotic outbreaks may be related to environmental and socio-economic factors that could affect dead-end host, vector and reservoir species [16]. Pathogeographic models allow the needed multi-level understanding of the temporal and spatial patterns of infectious diseases and, consequently, are useful for predicting the risk of WNF outbreaks [17]. Given that environmental data are continuous and no clear thresholds can be established to predict a zoonotic disease, fuzzy logic provides a suitable approach for modelling the relationships between disease outbreaks and environmental data [18]. In fact, fuzzy logic [19] has been already applied to the development of predictive models related to WNV at a biochemical level [20].

Here we used fuzzy logic, through the application of favorability functions [21], to assess to what degree certain environmental conditions or particular geographical locations can favor the emergence of WNF, even in places where it has not yet been reported. The ultimate aim of this study was to produce a cartographic model of the risk of occurrence of WNF cases in

Europe, which can be useful for the prevention of and early response to future outbreaks. Specifically, we analyzed the geographic variation of the cases recorded in 2017 to model the environmental and spatial factors that determine the future risk of the disease incidence, in terms of its occurrence and prevalence. Then we tested the capacity of the risk models to predict, both in space and time, the apparently unusual pattern of WNV disease cases during the 2018 transmission season.

## Methods

### Study area and distribution data of WNF

The study area consisted of the European continent except for Russia, Belarus, Ukraine and Moldova. The total area of study was divided into NUTS (Nomenclature of Territorial Units for Statistics) level 3 [22], which were used as operational geographic units (OGUs [23]) (Fig 1). However, for Belgium, the Netherlands, Germany and Switzerland, the NUTS level 2 were used to increase homogeneity in the size of the OGUs. Similarly, the different NUTS that make up the London area (Outer London—East and North East; Outer London—West and North West; Outer London—South; Inner London—West and Inner London—East) were joined together as a single OGU: London. This resulted in a total of 931 OGUs (Fig 1).

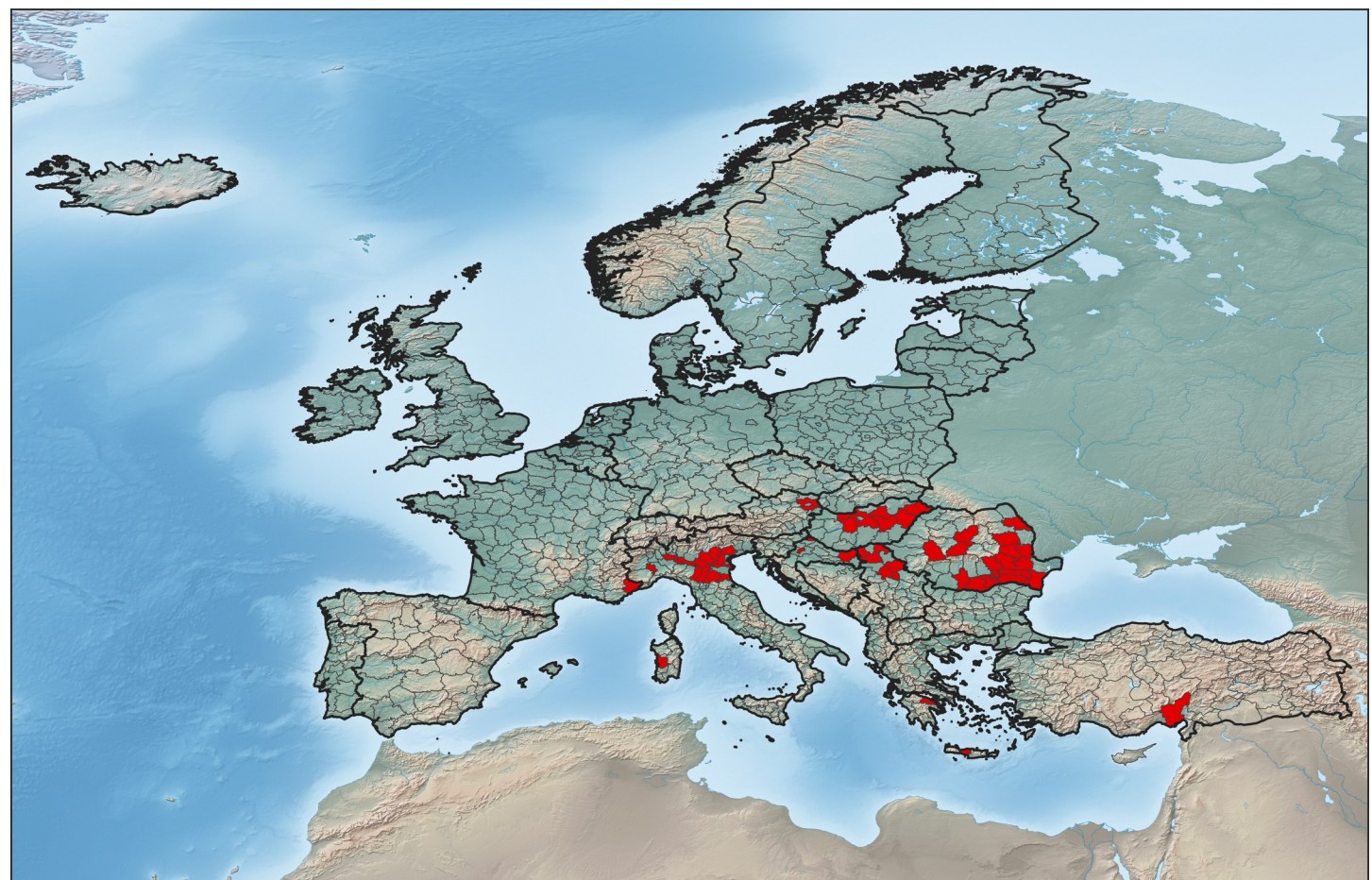

**Fig 1. Study area. Countries involved in the study area (black contour line) and their corresponding OGUs (grey contour line).** OGUs with confirmed cases of WNF in 2017 are shown in red.

We got epidemiological data from the European Center for Disease Control and Prevention (ECDC, 2019) for 2017 and 2018. Only confirmed cases of WNF were used following at least one of the European criteria for confirmed cases [24]: a) isolation of WNV from blood or cerebrospinal fluid (CSF); b) detection of WNV nucleic acid in blood or CSF; c) WNV-specific antibody response (immunoglobulin M; IgM) in CSF; or d) WNV IgM high titer, detection of WNV IgG and confirmation by serum neutralization test. With this information, we obtained a binary variable defining the presence or absence of cases in each OGU in 2017 (Fig 1). We also obtained a binary variable for the presence or absence of cases in each OGU in 2018, and another variable with the number of cases in each OGU in 2018, to validate the risk-models predictions regarding outbreak location and intensity, respectively (see below). The temporal pattern of occurrence of WNF cases in 2018 was measured using the Julian week of the first case recorded in the OGUs, which was used to validate the temporal predictions of the risk models produced with information from 2017 (see below).

### Predictive variables

The set of environmental variables used to describe, explain and predict the spatio-temporal pattern of occurrence of WNF cases in the study area is shown in S1 Table. These variables were selected because they were assumed to have a potential effect on life cycles and body conditions of mosquitoes, birds or humans, or to be at least correlated with more proximal causal factors. In addition, we used the geographical coordinates, latitude and longitude, of the geographical barycenter of each OGU to assess the spatial structure of the distribution pattern of occurrence of WNF cases [18]. All these variables were used to build the predictive risk models based on the information recorded in 2017.

### Modelling procedure

We used biogeographical modelling based on fuzzy logic to analyze, separately, the environmental envelope and the spatial structure of the disease in the study area. The disease environmental envelope model was produced following several steps. First, we assessed the individual explanatory power of every environmental variable (S1 Table). This power was established according to the significance of score tests in univariate logistic regressions of the presence/absence of 2017 cases in each OGU on each environmental variable separately [25]. In order to control the noise effect of multicollinearity among the environmental variables, pairwise Spearman correlation coefficients were calculated between all the variables. If two variables had a correlation higher than 0.8 and were included in the same subtype (see S1 Table), the variable with less individual explanatory power was deleted from the subsequent steps.

To limit the increase in type I error derived from the number of environmental variables considered, we controlled the False Discovery Rate (FDR) [26]. To do this, the remaining environmental variables were arranged in decreasing order according to their significance for explaining the distribution of disease cases. Only the variables whose significance in the score test was less than $i{\times}q/V$ (where $i$ is the position of the variable in the order referred, $q = 0.05$ was the FDR, and $V$ was the total number of remaining variables) were used in the following steps.

A multivariate logistic regression model was then produced via a step-by-step introduction of the subset of variables that went over the previous filters. This stepwise process started with a null model with no explanatory variable, and then a variable was added at each step only if the resulting new regression was significantly improved compared to that produced in the previous step according to the omnibus test. Thus, the environmental variables included in the final environmental model were a parsimonious representation of all the effects imputable to

the set of variables analyzed in the stepwise procedure. The values of the parameters of the logistic regression were established by maximum likelihood estimation using a machine learning algorithm. The weight of every variable in the model was assessed with the test of Wald.

The result of the multivariate logistic regression was, in each OGU, a probability value ($P$) of presenting at least one human case of WNF based on the environmental conditions. $P$ values were transformed into favorability ($F$) values using the Favorability Function [21]:

$$F = \frac{\frac{P}{1-P}}{\left(\frac{n1}{n0}\right) + \left(\frac{P}{1-P}\right)}$$

$n1$ is the number of OGUs with reported cases, and $n0$ the number of OGUs with no case reported. In this way, an environmental $F$ value (ranging from 0 to 1) was calculated for each OGU, which represents the degree to which environmental conditions at that OGU favor the occurrence of WNF outbreaks.

We also produced a spatio-environmental WNF model. Firstly, we analyzed the spatial structure of the disease in the study area in 2017, which may be attributed to contagious biological processes that affect the progression of the disease [27] such as, for example, the chronology of arrivals of the reservoir species or the dispersion of hosts and vectors. For that purpose, a purely spatial variable was constructed based on the coordinates latitude ($La$) and longitude ($Lo$). A polynomial trend-surface analysis [28] was conducted through a logistic regression of the presence/absence of cases in the OGUs on nine different combinations of latitude and longitude (i.e., $La$, $Lo$, $La^2$, $Lo^2$, $La$ x $Lo$, $La^3$, $Lo^3$, $La^2$ x $Lo$, $La$ x $Lo^2$), in backward steps. This produced a probability value of presenting at least one WNF case for each OGU according to its spatial location. Analogously to the environmental model, we obtained the spatial favorability for the occurrence of cases in the study area.

We used the resulting logit function of the spatial logistic regression as a new variable ($Ysp$), which was included together with the subset of variables selected in the multivariate environmental model to elaborate, through a new forward stepwise logistic regression, a spatio-environmental model, which took into account both the spatial structure and the environmental envelope of the WNF cases in 2017. All modelling processes were run with the IBM SPSS Statistics 25 software.

## Evaluation of the model descriptive capacity

We evaluated the discrimination and classification capacity of the models as well as their calibration with the data of 2017. The discrimination power of the models was assessed using the area under the receiver operating characteristic curve (AUC) [29]. The classification power, using the value of F = 0.5 as classification threshold, was estimated through the sensitivity, specificity, Cohen's kappa and correct classification rate (CCR) [30] and the over-prediction and under-prediction rates [31]. The calibration of the models was tested using the Hosmer and Lemeshow test [32], which is frequently used in risk prediction models. This test assessed whether the frequency of observed OGUs with WNF cases matched the expected frequency in subgroups of the OGUs, where subgroups were identified as the deciles of fitted risk probability values. Models for which differences between expected and observed numbers in subgroups were not significant were considered to be well-calibrated.

## Evaluation of the model predictive capacity

We tested if the models based on 2017 cases had the capacity to predict the overwhelming outbreak recorded in 2018, as follows.

**Predicting the occurrence of cases.** We tested whether the favorability for the presence of disease cases in each OGU in 2017 could be used to anticipate which OGUs were more likely to present cases of WNF during the following year. We hypothesized that the spatio-environmental model should be more useful than the purely environmental model for this aim since both the environmental and the spatial characteristics associated to the OGUs should be related with the expected location of cases. In any case, we tested the predictive classification and discrimination capacity of both the environmental and the spatio-environmental models, as well as their calibration, with information obtained in 2018. The discrimination power of the models was assessed using the AUC [29], whereas the classification power was estimated through the sensitivity, specificity, kappa, CCR, and the over-prediction and under-prediction rates, using the value of F = 0.5 as classification threshold [31]. Hosmer and Lemeshow's test cannot be used to validate the calibration of the predictive models when the prevalence in the values to be predicted differ from that in the training dataset, as prevalence affects probability values [21]. The calibration of the predictions was assessed instead using Miller's calibration line [33], which assesses the bias and spread of predictions. Miller's line has the intercept and slope of the logit of a logistic regression of observations on the logit of predicted probabilities. Therefore, this is of no use with the training data, as perfect calibration (i.e., intercept = 0 and slope = 1) is always attained on the same data that were used for building the model, but it is useful for validating a model when applied to predict other data, such as those of 2018 (Miller et al., 1991). Intercept values different from 0 indicate variation in overall prevalence, whereas slope values are indicative of the calibration of the trend in the predictions.

**Predicting the intensity of cases.** We hypothesized that, once WNV has reached a location, the environmental model should be a better predictor of the number of cases per OGU than the spatio-environmental model. This hypothesis is based on the assumption that it is the environmental potential of each region for the development of WNV infections that should favor the internal spread of the disease once the outbreak has started. Nevertheless, in those OGUs with WNF cases during 2018, we tested if there was a correlation between the number of cases in 2018 and both the environmental and the spatio-environmental favorability values, using Spearman's correlation coefficient. A positive relationship between environmental favorability and abundance of species has already been demonstrated [34] but, to our knowledge, such a relationship has never been confirmed for the incidence of any zoonotic disease.

**Predicting the imminence of cases.** We tested the hypothesis that the spatio-environmental favorability of an OGU for the occurrence of the disease could anticipate how early disease cases would occur in the course of the following year. We hypothesised that the higher the spatial and environmental favorability, the earlier the cases would occur. We tested the correlation between the Julian week of the first case of the disease in 2018 and both the environmental and the spatio-environmental favorability (as derived from 2017 data) for the occurrence of disease cases, using Spearman's correlation coefficient.

As the development of the disease occurs through a contagious process highly dependent on the location of the first case, we first grouped the OGUs according to countries to assess the correlation within each country. Furthermore, we also tested if the temporal spread of the disease could be more associated to ecological systems, such as river basins, than to countries, as basins are natural geographic units with well-defined orographic limits and are known to affect the distribution of water-dependent organisms, such as mosquitoes, and birds [35–37]. Given the magnitude of the hydrographic basin of the Danube, we tested this hypothesis comparing: on the one hand, the correlation between the environmental favorability and the Julian week of the first case in the OGUs comprised by the entire basin of the Danube, irrespective of the country they belong to; and, on the other hand, the same correlation separately for the OGUs belonging to each of the countries totally or partially overlapping with the Danube basin.

## Results

### Environmental model

The occurrence of WNF cases was favored at OGUs with natural characteristics such as low mean altitude, high temperature in the warmest month, high annual temperature range, and presence of river courses (S2 Table). Rain-fed agriculture, as well as poultry and horse livestock were the human activities that also contributed to explain the distribution of WNF (S2 Table). The favorable areas in Europe for the presence of WNF cases, according to the environmental envelope, are shown in Fig 2, with favorability values grouped in four ranges: low (0–0.2), intermediate-low (0.2–0.5), intermediate-high (0.5–0.8) and high (0.8–1). The highest values were located in OGUs that reported cases in 2017, such as the Danube Basin (involving Romania, Bulgaria, Serbia and Hungary), the Po Basin in the north of Italy, the Greek OGUs around the Aegean Sea, and the south of Turkey (compare Figs 1 and 2). However, other OGUs with no cases in 2017 also showed high or medium-high environmental favorability for the occurrence of WNF, e.g., Poland, the Netherland or the south of Spain (Fig 2).

### Spatial model

The logit of the spatial structure ($Ysp$) of the disease is shown in S3 Table. The values of the Wald test show that latitude has a higher weight than longitude in this spatial structure, although longitude also has a role. The spatially favorable OGUs in the study area for the presence of WNF cases are shown in Fig 3. There are two zones with high spatial favorability for the presence of cases in the study area, namely the southeast of the Mediterranean Sea,

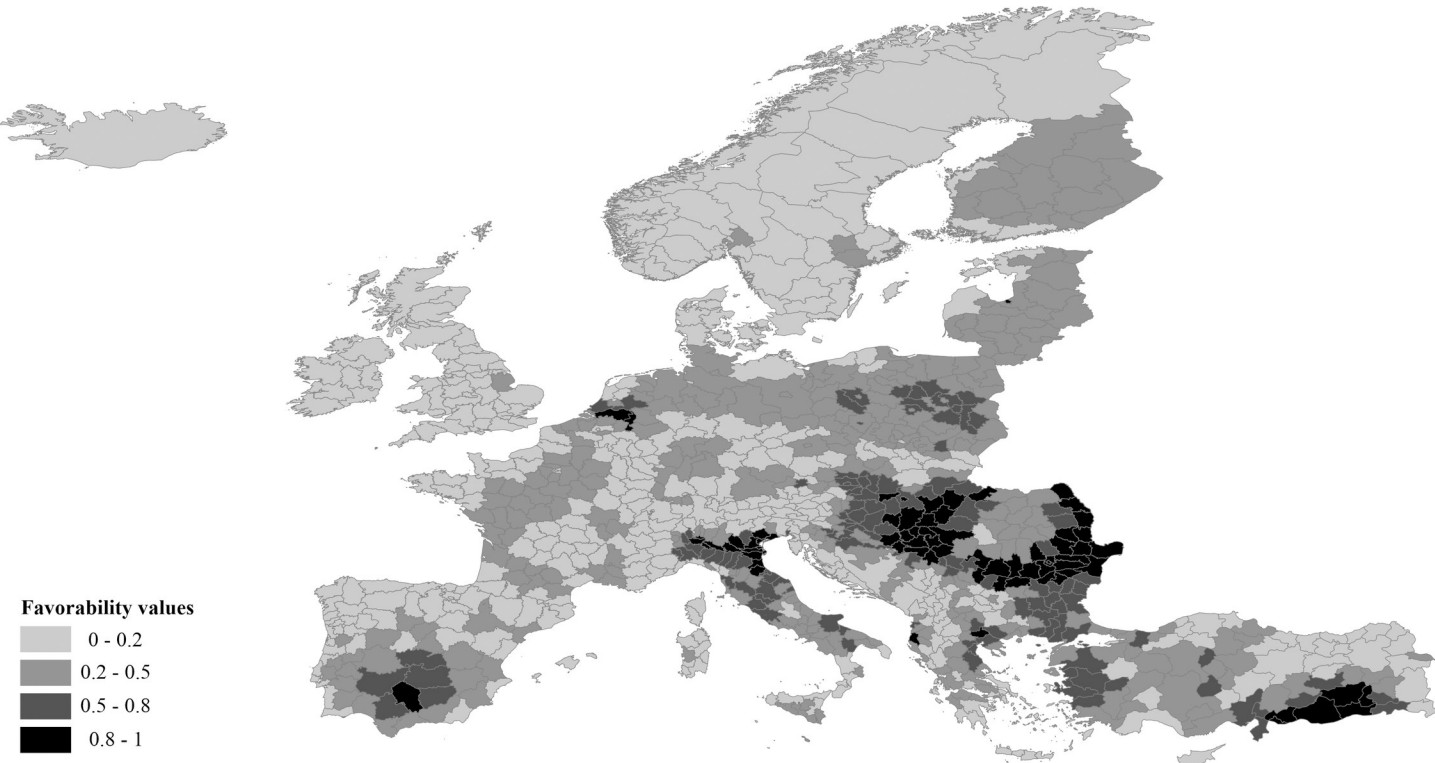

**Favorability values**
- 0 - 0.2
- 0.2 - 0.5
- 0.5 - 0.8
- 0.8 - 1

**Fig 2. Cartographic model of the environmental favorability for WNV infection in humans in the study area.** It results from the projection of the mathematical model shown in S2 Table, which is based on cases reported in 2017.

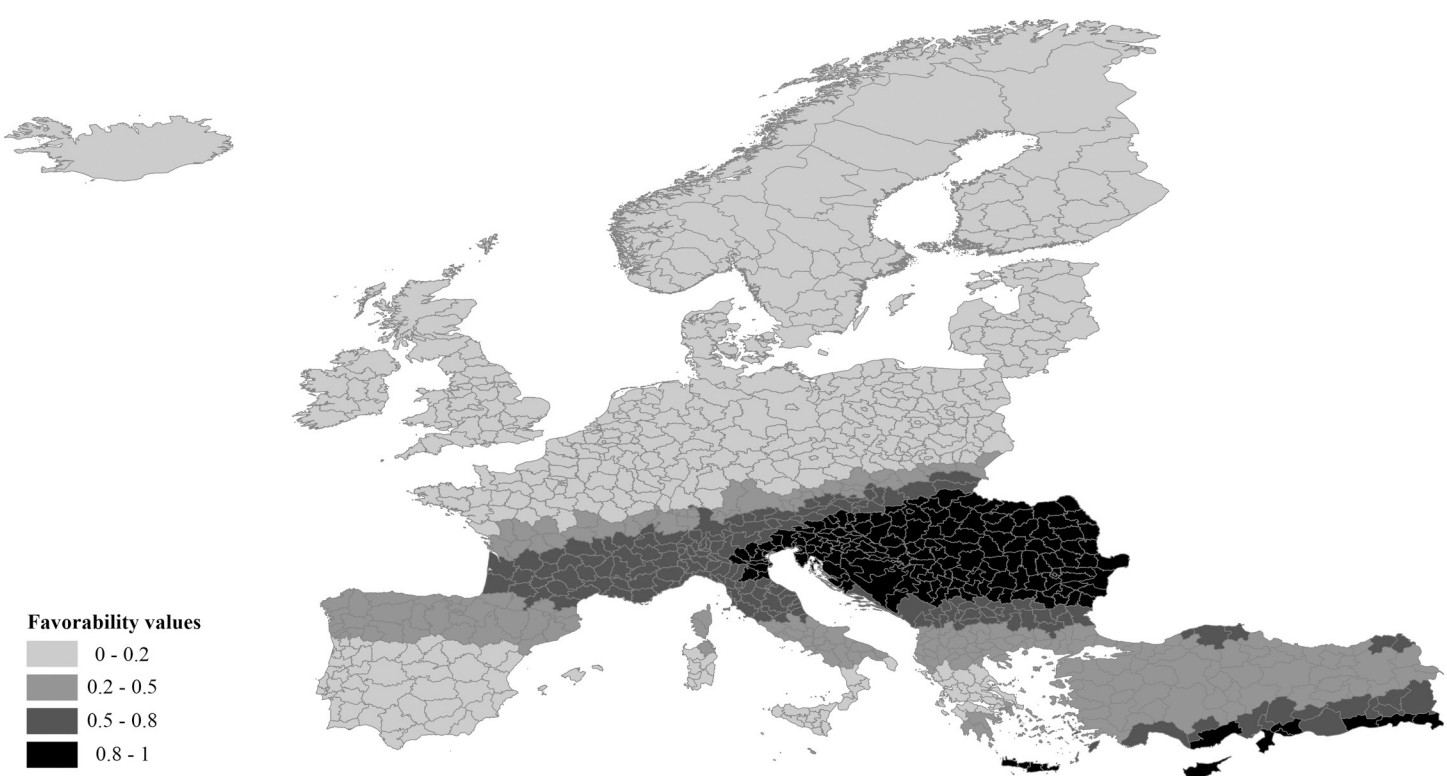

**Fig 3. Cartographic model of the spatial favorability for WNV infection in humans in the study area.** It results from the projection of the mathematical model shown in S3 Table, which is based on cases of 2017.

comprising southern Greece and the Anatolian Peninsula, and the main nucleus of the spatial structure, which occupies the entire Danube river basin and the eastern part of the Po basin (Fig 3).

## Spatio-environmental model

According to the spatio-environmental model, the occurrence of WNF cases is favored by a combination of environmental variables and the spatial structure of the disease. In comparison with the purely environmental model, the spatial structure ($Ysp$) was added instead of altitude, annual temperature range and horse density (S4 Table). The high and medium-high favorable areas were mostly located in the south-east of Europe, with the main core in the basin of the Danube River (Fig 4A).

An assessment of the capacity of the environmental and the spatio-environmental models to correctly classify and discriminate WNF cases reported in 2017 can be seen in Table 1. The spatio-environmental model had a greater classification capacity than the solely environmental model, with greater kappa, sensitivity, specificity and CCR values and lower under-prediction and over-prediction rates (Table 1). Discrimination capacity was also higher in the spatio-environmental model than in the environmental model (Table 1), being excellent for the environmental model (>0.80) and outstanding for the spatio-environmental model (>0.90) [38]. The calibration of both models was good, as the Hosmer and Lemeshow test showed that differences between expected and observed cases were not significant for both models (environmental model HL = 11.32; $p$>0.05; spatio-environmental model HL = 7.85; $p$>0.05).

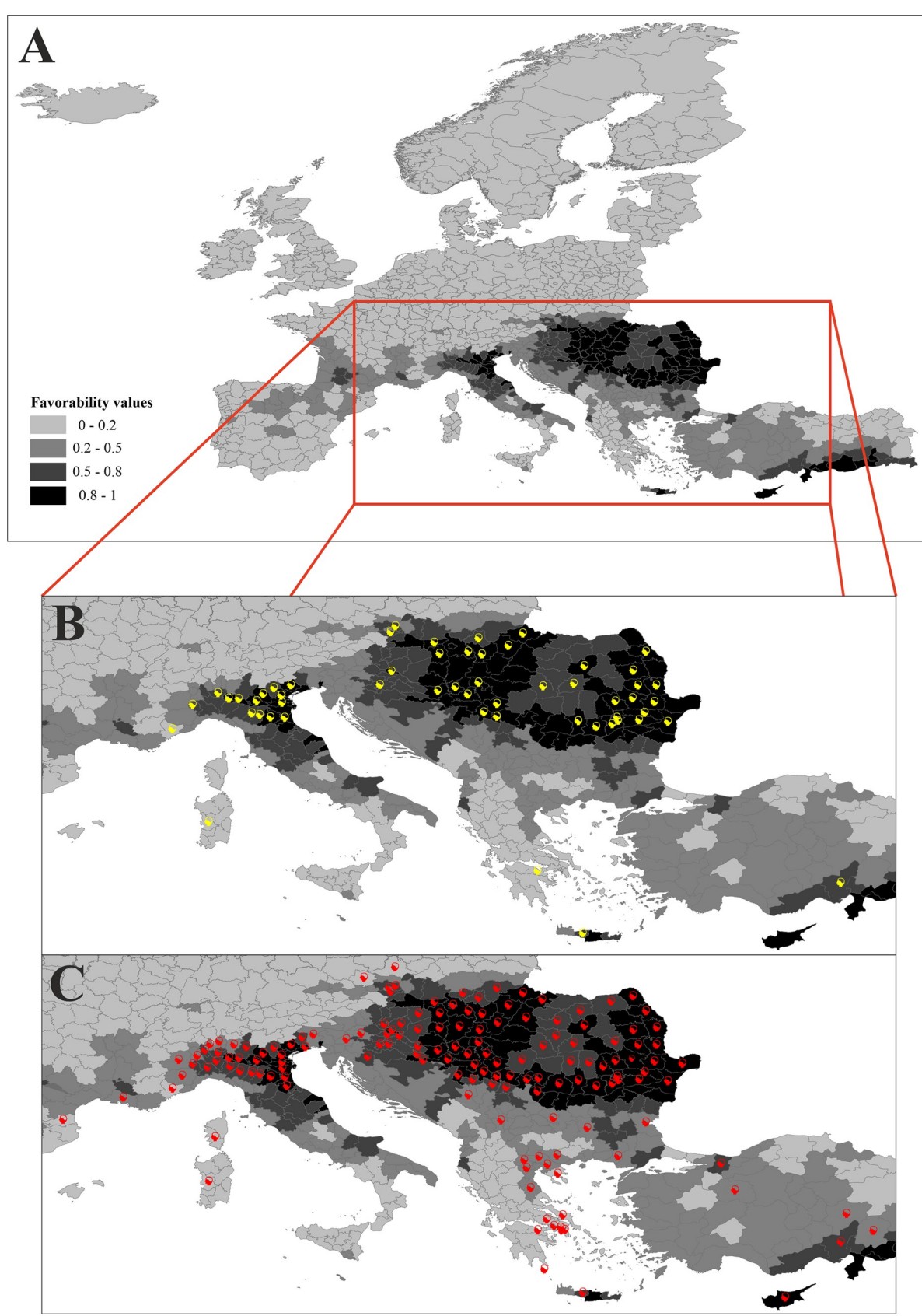

Favorability values
- 0 - 0.2
- 0.2 - 0.5
- 0.5 - 0.8
- 0.8 - 1

**Fig 4. Spatio-environmental model of favorability for the occurrence of WNF cases in humans.** (A) Darker OGUs show higher spatio-environmental favorability. (B) Yellow icons indicate those OGUs that presented cases in 2017. (C) Red icons indicate the OGUs that presented cases in 2018.

## Prediction of cases: Where, how many and when

**Predicting the occurrence of cases.** The number of OGUs with confirmed cases increased from 55 in 2017 to 151 in 2018. Following our hypothesis, the spatio-environmental model had a greater predictive classification and discrimination capacity over the OGUS with detected outbreaks in 2018 than the environmental model (Table 1). In fact, the spatio-environmental model had an outstanding predictive discrimination capacity (>0.90) [38]. Remarkably, the models, both environmental and spatio-environmental, had higher kappa, specificity and CCR values as well as lower over-prediction rates when evaluated with the occurrences of the following year (2018) than with the cases with which they were built (2017). The Miller calibration lines for 2018 (Fig 5) had slopes slightly lower than 1 and intercepts higher than 0. The intercept higher than 0 implies that the models could not predict the increase in overall probability of occurrence, i.e., the higher prevalence of the disease in 2018 was due to causes not included in the models. The slopes close to 1 indicate that the models succeeded in predicting the relative trend of the probability pattern, i. e., the probability gradient throughout the OGUs was well predicted by the models.

Almost 90% of the OGUs with cases detected in 2018 occurred in the 40% of OGUs, whose spatio-environmental favorability was at least intermediate (F>0.2). Although only 20% of the study area had spatio-environmental favorability values higher than 0.5, 71% of the OGUs with cases in 2018 had these values (Fig 4). Most significantly, more than 42% of the OGUs with cases in 2018 were among the less than 10% of OGUs with high spatio-environmental favorability (F>0.8).

**Predicting the intensity of cases.** The number of human cases detected in Europe jumped from 197 in 2017 to 1,605 in 2018. The number of cases per OGU reported in 2018 was more correlated with the environmental favorability (Spearman correlation = 0.35; $p$ = 1.2E-5) than with the spatio-environmental favorability (Spearman correlation = 0.26; $p$ = 1.06E-3). This agrees with our hypothesis that, once WNV has reached an area, the spatial structure of the disease is of less importance than the environmental characteristics of the OGU for determining the intensity of infections. The OGUs with the highest number of cases tended to have values of environmental favorability higher than 0.8 (Fig 6). The OGUs with high environmental favorability (F>0.8) enclosed 60% of the total number of cases reported in 2018, and an additional 25% of cases were in OGUs with intermediate-high environmental favorability (0.5<F<0.8).

**Table 1. Comparative assessment of the classification and discrimination capacities of models.** Values for the environmental and spatio-environmental models, considering the disease case reports of 2017 and 2018. We used the favorability value 0.5 as a cut-off point for classification purposes.

|  | 2017 occurrences | | 2018 occurrences | |
|---|---|---|---|---|
|  | **Environmental model** | **Spatio-environmental model** | **Environmental model** | **Spatio-environmental model** |
| **Classification** |  |  |  |  |
| *Kappa* | 0,3229 | 0,3905 | 0,4579 | 0,5551 |
| *Sensitivity* | 0,8095 | 0,9048 | 0,6424 | 0,7152 |
| *Specificity* | 0,8318 | 0,8479 | 0,8718 | 0,8962 |
| *CCR* | 0,8303 | 0,8518 | 0,8346 | 0,8668 |
| *Underprediction* | 0,0163 | 0,0081 | 0,0736 | 0,0580 |
| *Overprediction* | 0,7411 | 0,6984 | 0,5076 | 0,4286 |
| **Discrimination** |  |  |  |  |
| *AUC* | 0,8729 | 0,9275 | 0,8167 | 0,9067 |

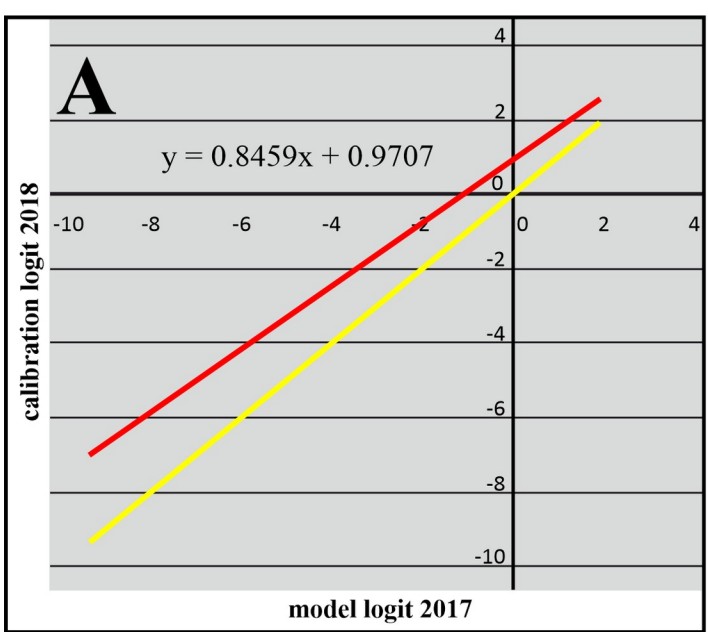

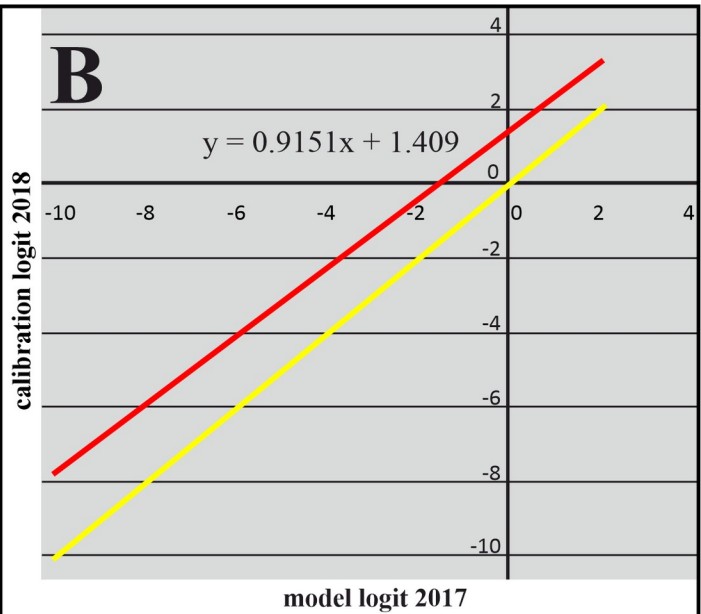

**Fig 5. Predictive Miller calibration lines for the environmental (A) and spatio-environmental (B) models.** The red calibration line is the graphic representation of the calibration logit, which is the logit of a logistic regression of the WNF occurrences of the year 2018 on the model's logit values (Y-axis), along the model's logit values (X-axis). The equation of the red line is shown. The yellow line represents the perfect calibration line, with slope 1 and intercept value 0, for comparison.

**Predicting the imminence of cases.** Contrary to our expectations, the earliness of WNF cases during 2018 was more significantly correlated with the environmental favorability (Spearman correlation = -0.35; $p$ = 1.4E-5) than with the spatio-environmental favorability (Spearman correlation = -0.26; $p$ = 1.2E-3). Specifically, the Julian week in which the first case occurred was earlier in the OGUs that showed the highest environmental favorability values. This relationship between time of the first reported case and environmental favorability was higher when river basins, rather than countries, were considered as a context. The OGUs inside the Danube river basin showed the most significant negative correlation between the

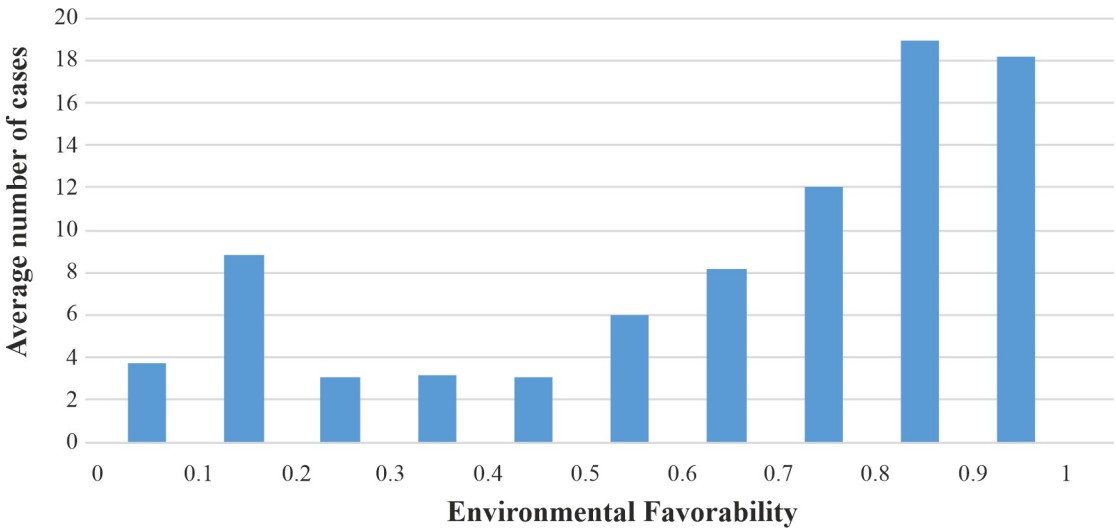

**Fig 6. Mean of 2018 WNF cases per OGU for each environmental favorability interval.**

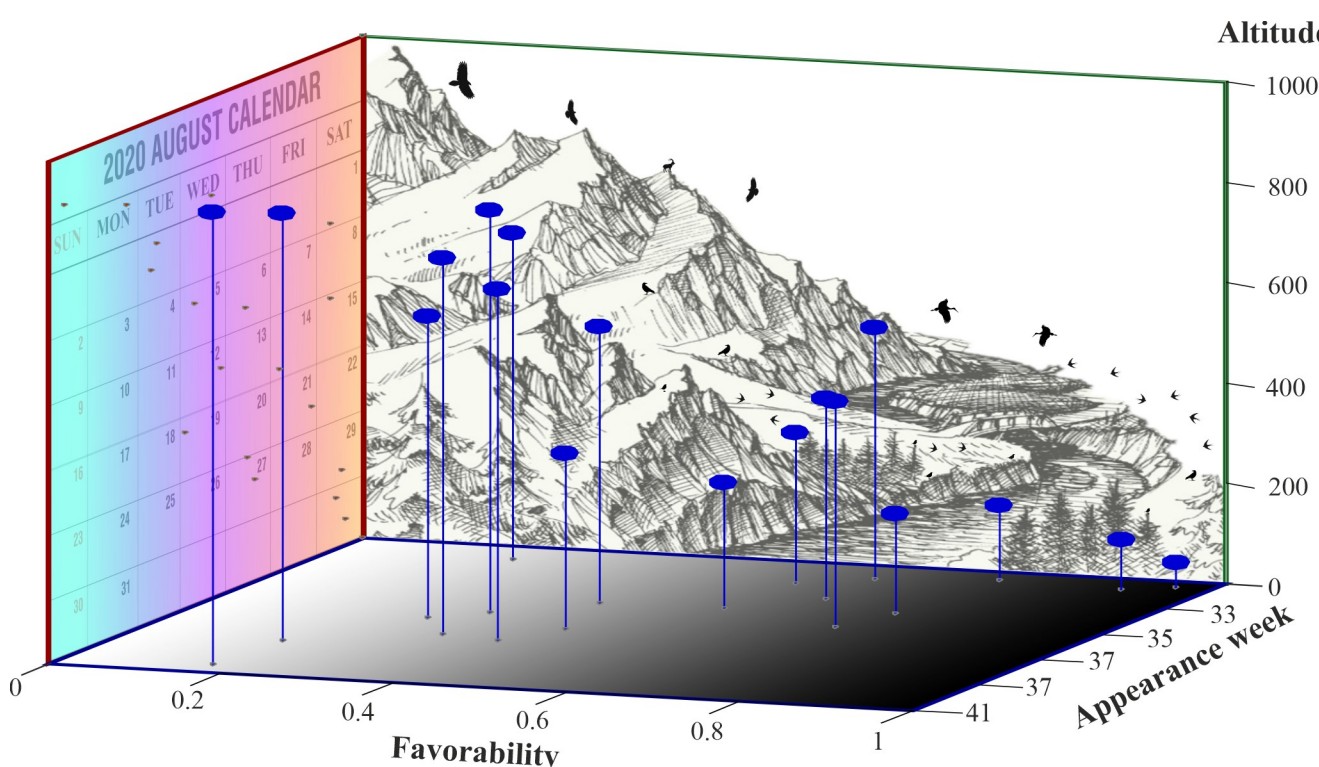

**Fig 7. 3D representation of the average values of environmental favorability (X-axis), Julian week of appearance of the first case of disease (Z-axis), and altitude in 50-meter altitude ranges (Y-axis) of the OGUs.** The color gradient (from white to black) in the X-Z plane reflects the degree of environmental favorability of the OGUs (from low to high). The color gradient (from orange to light blue) in the Y-Z plane reflects the earliness of the disease onset (earlier to later).

time of appearance of the first cases and the environmental favorability (Spearman correlation = -0.52; *p* = 2.23E-7). However, when the correlations were performed within each country inside this river basin (i.e. Croatia, Hungary, Romania, Serbia, Austria and Bulgaria), the mean correlation value was much lower (-0.12), and only Romania and Serbia showed significant correlations between favorability and the first occurrence time, with correlation values of -0.54 (*p* = 9.60E-4) and -0.75 (*p* = 8.01E-4), respectively. This suggests that river basins, rather than countries, had the main role in the temporal spread of the disease. This notion is also supported by the fact that most cases within Italy occurred in the OGUs located within the Po river basin (Fig 4). Furthermore, the Julian week of the first case was correlated with the number of cases per OGU (Spearman r = -0.643, p = 5.802E-19).

A low altitude was significantly related with the environmental favorability for the occurrence of WNF cases (see S2 Table) and, thus, was related with the earliness of cases in the OGUs. However, the role of altitude could be also linked to the relevance of river basins as units of temporal spread, given that basins are typically delimited by mountains. The Julian week of the first cases in 2018 was, in fact, more significantly correlated with the average altitude of the OGUs (Spearman correlation = 0.39, *p* = 5.64E-7) than with the overall environmental favorability; i.e., the OGUs with a higher altitude tended particularly to present later cases along the propagation season than those located at lower elevations. This suggests that mountains could be barriers, or at least filters, for the spread of the disease. To clarify this complex interrelation, we showed the temporal spread of disease cases concerning environmental favorability and elevation in Fig 7.

## Discussion

### Environmental risk model

Our environmental risk model for WNF (Fig 2 and S2 Table) was quite parsimonious and straightforward, including only seven predictors. Although certain topographical and hydrographic characteristics of the OGUs, as well as some human activities related to agriculture and livestock practices, can favor the occurrence of outbreaks of WNF, climate was the most important factor, as it provided the two most significant predictors of the environmental risk (S1 and S2 Tables). Human density and infrastructures or types of ecosystems did not add significant explanatory power to the model. Precipitation was also non-explanatory, perhaps because potentially more influential features such as precipitation in spring and early summer were not available in the CHELSA database.

**Human-related predictors.** Rain-fed crops contributed significantly to favor the distribution of WNV infections. Other studies also showed that the incidence of WNF had a positive relationship with agriculture [39]. It has been hypothesized that this relationship could be due to differences in the mosquito communities inhabiting the urban and agricultural environments [40]. However, host diversity could be also behind this association. The relationship between biodiversity and transmission of infectious diseases is known [41]. A high diversity of birds has been shown to have a negative effect on the prevalence of WNF, acting as a natural buffer to WNV amplification [42]. This is so because, as the local diversity of bird species increases, the abundance of non-competent hosts also increases, which entails a reduction in the rate of encounters between infected hosts and vectors. The intensification of agriculture is known to contribute to a reduction in biodiversity [43], agricultural areas being usually linked to a low diversity of birds [44]. Therefore, rain-fed agricultural areas could favor the distribution of WNF cases in Europe by decreasing the local diversity of birds, which prevents the dilution of disease transmission between vectors and competent hosts.

Poultry farms seem to favor the occurrence of WNF cases in the OGUs (S2 Table). In the same way as other birds, hens and chickens (*Gallus gallus*) are susceptible to get infected by WNV [45]. Therefore, the presence of poultry farms may be a potential source of virus amplification. Poultry farms would facilitate the establishment of the virus brought by migrating birds. Consequently, a higher number of poultry farms would facilitate the bird-to-bird transmission by mosquitoes, thus amplifying the enzootic cycle, with farms acting as efficient reservoirs of the disease and facilitating transmission to people.

Horse density also appears to favor the outbreak of the disease. This may be caused by the fact that equestrian farms provide suitable environments and breeding sites for the proliferation of mosquitoes, water troughs, ponds and purines being the most favorable habitats for the development of mosquito larvae [46]. *C. pipiens* is the principal vector of WNV in Europe [3] and the most abundant mosquito species, by far, in equestrian farms [46]. In fact, in 2018 WNV was for the first time detected in Germany among horses from the Brandenburg and Sachsen-Anhalt region. The following year, WNV was first detected among humans in the country, in the region of Sachsen-Anhalt and in the contiguous regions of Leipzig and Berlin [14]. Thus, horses may not only be dead-end hosts in the WNV cycle, but their farms may also act as a focus of proliferation of its main vector.

**The role of temperature.** The most significant environmental predictor of WNF was the increase in temperature annual range. This could be related to the diversity and abundance of migrant breeding bird species. A higher temperature range implies a higher seasonality in environmental energy and, thus, in the availability of food resources. This may be detrimental for resident bird species, as food availability is less stable, but, precisely because of that, might be favorable for migratory breeding species. A high-temperature range increases the number

of summer migrants waterbird species in the European river basins by reducing the number of competing resident species [36]. Therefore, our results suggest that WNV-carrying migratory birds may be attracted by the periodic abundance of food and habitat that competing resident species could not fully utilize.

According to our results, the maximum temperature of the warmest month was the second most significant variable for explaining the distribution of cases of WNF in Europe (S2 and S4 Tables). The weather has been previously proven to be related to the prevalence of WNF [47], especially high temperatures, which are one of the most important drivers for WNV transmission [48]. WNV circulation in northern Europe is probably limited by low temperatures [48]. Temperature has a significant effect on the virus and the vectors, as high values increase replication rates [49], and make mosquitoes transmit the virus earlier by shortening the gonotrophic cycle and increasing the bite rate [50].

This is why climate warming may produce changes in the expansion of vectors, both at higher altitudes and at higher latitudes [51], favoring the establishment of WNV in new regions [52]. Climate warming is increasing the frequency and severity of extreme weather events, such as heatwaves, floods or droughts, thus intensifying the interactions between vectors, viruses and hosts [52]. Droughts, for example, increase the abundance of mosquitoes, ultimately favoring the risk of transmission of WNV to humans [47].

**The effect of topo-hydrographic features.** High elevations hinder the occurrence of WNF cases (S2 Table). High altitude has been previously proposed as an impediment that may delimit the expansion of disease vectors, including mosquitoes, due to climatic factors, such as the decrease in temperature and humidity [51]. In Greece, a link between WNF cases and low altitude was found, which was attributed to a negative relation between vector density and elevation [53]. River courses in the OGUs favored the occurrence of cases (S2 Table), which could be related with water availability, which is an important factor in the creation of favorable habitats for larvae and mosquitoes [54].

## Spatial risk model

The spatial model (Fig 3) is in accordance with other studies that reported a high prevalence of WNF within the Danube and Po basins [12,55]. The Danube basin constitutes an important settlement area for migratory birds from Europe, Asia and Africa, and consequently a concentration area for the pathogens carried by migrant birds during their routes along the stopover and feeding areas [56].

For species breeding in Europe and wintering in Africa, there is a strong relationship between the longitudes of breeding and non-breeding sites [57]. Therefore, migratory birds that overwinter in East Africa mostly use the eastern migration route, either crossing the Danube Basin or breeding in it [58]. Passerines, in particular, basically follow two broad routes in their Afro-Palearctic migration: the western and the eastern one [59]. Given the formidable ecological barrier posed by the Sahara Desert, the eastern migration route is predominantly driven along the eastern edge, where the longitudinal extent of the hospitable area is reduced [59]. Therefore, the Nile River acts as a corridor connecting sub-Saharan Africa with the Palearctic for migratory birds. This may be the reason for the existence of a spatially favorable area for WNF in the Anatolian coast, facing the mouth of the Nile (Fig 3). Indeed, countries along the Nile Basin (i.e. Egypt, Sudan, South Sudan and Uganda) show a high prevalence of the disease [60,61].

## Prediction of cases: Occurrence, intensity and imminence

The fuzzy concept of favorability allowed us to identify the effects of the intrinsic conditions of the OGUs on the distribution of WNF independently of the prevalence of the disease (i.e. of

the absolute incidence of WNF cases) [21]. Thanks to this, not only the risk for occurrence of the disease was predicted for the following year, but also the distribution of the number of cases per OGU that would occur in the next transmission season, as well as the imminence/ timing of disease propagation.

The unusually intense outbreak that happened in Europe in 2018 could be a symptom of a new phase of WNV expansion [62]. It could be a consequence of the atypically hot summer of 2018 [63]. The variable temperature in the warmest month we used in our environmental model was an average of 35 years (1979–2013), which prevented the model from predicting the overall higher prevalence of the disease in 2018. However, the general trend was well predicted, as temperatures were uniformly higher in 2018 than in 2017, so local prevalence was higher, but the differential pattern among OGUSs was maintained.

The predictive spatio-environmental model was more effective than the exclusively environmental model when classifying and discriminating the OGUs with cases of WNF, both in 2017 and in 2018. The spatio-environmental model considered the potential effects of biotic contagious processes on the distribution of outbreaks, such as the phenology and the routes of migration of the reservoir species, or the dispersion of hosts and vectors. Our results suggest that these processes may restrict the disease to southern and eastern Europe. Therefore, to update the risk map of WNF outbreaks in Europe it is important to efficiently update the spatial distribution of cases.

On the other hand, the environmental model highlights the potential of non-affected areas to develop disease cases if the spatial trend of the disease changes. For example, although WNF cases were not present in Spain in 2017, the environmental model detected favorable OGUs for the disease in southern Spain, where cases and deaths recently occurred [64]. Inside the affected areas, the environmental model was able to predict the number of cases recorded in the different OGUs. In other words, our environmental model can predict how intensely the cases will occur in the OGUs. This could be linked to the fact that environmental favorability values were also related to the imminence of disease cases in the following year. The greater the environmental favorability of a territory, the earlier the virus manifests, and the more time it has for developing and accumulating cases until the end of the season. This was corroborated by the significant correlation between the number of cases per OGU and the Julian week of the first case.

## The role of river basins

Our results highlight the importance of river basins as biogeographic units in the disease propagation. An excellent example is the case of Italy, where all the 31 continental OGUs that presented cases in 2018 were completely within the Po river basin, independently of the political unit they belong to (Fig 4). River basins act as natural geographical units with ecological functions [36,37].

Rivers, lakes and wetlands are linked through water movements within river basins [36,37] and, thus, are normally affected by similar biogeographical and ecological factors within the same basin, which are diverse in different basins. Birds can pass over boundaries of river watersheds; however, their movements, apart from migrations, predominantly occurs inside the river basin [37]. That is also true for mosquitoes, as water availability for their breeding is shared within a river basin. Therefore, if river basins affect vectors and reservoir of the virus, the virus could spread more easily within the same watershed.

The highest parts of river basins are the most unfavorable ones for the presence of WNF, and they are the last to manifest cases. The first cases were reported in low regions, and these were followed by cases in medium-altitude areas, the most elevated OGUs being those that

presented the least and latest cases within the river basin. Following this reasoning, the spread of the disease would begin in the lowest areas of the valleys, with the greatest water availability and the highest density of vectors and reservoirs. From there, it would spread gradually throughout the entire river basin occupying firstly the most favorable low-altitude areas and then to the highest and least favorable ones. Therefore, country-specific actions would be insufficient for managing the yearly dynamic of WNF spread, since it may be conditioned by environmental and ecological characteristics that affect all countries within a basin. Thus, decision-making would be more efficient from an ecosystem perspective, considering river basins as operational units, irrespective of political borders. This, together with the awareness of the human and environmental factors that favor the appearance and propagation of WNF outbreaks, may lead to taking early actions in the high-risk areas that could reduce the impact of the disease and the number of affected people.

## Supporting information

**S1 Table. Explanatory variables used in WNV 2017 cases models in Europe.**
(DOCX)

**S2 Table. Explanatory variables included in the model of environmental favorability for the occurrence of WNF, based on cases of 2017.**
(DOCX)

**S3 Table. Combination of geographical coordinates included in the logit of the spatial favorability model for the occurrence of WNF, based on cases of 2017.**
(DOCX)

**S4 Table. Explanatory variables included in the spatio-environmental favorability model for the occurrence of WNF, based on cases of 2017.**
(DOCX)

## Author Contributions

**Conceptualization:** José-María García-Carrasco, Raimundo Real.

**Formal analysis:** José-María García-Carrasco.

**Investigation:** José-María García-Carrasco, Raimundo Real.

**Methodology:** José-María García-Carrasco, Jesús Olivero, Raimundo Real.

**Supervision:** Antonio-Román Muñoz, Jesús Olivero, Raimundo Real.

**Visualization:** José-María García-Carrasco.

**Writing – original draft:** José-María García-Carrasco.

**Writing – review & editing:** José-María García-Carrasco, Antonio-Román Muñoz, Jesús Olivero, Marina Segura, Raimundo Real.

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
