## [Decision Letter · Decision Letter 0]

24 Sep 2020

Dear Mr. García-Carrasco,

Thank you very much for submitting your manuscript "Predicting the spatio-temporal spread of the West Nile virus in Europe" for consideration at PLOS Neglected Tropical Diseases. As with all papers reviewed by the journal, your manuscript was reviewed by members of the editorial board and by several independent reviewers. The reviewers appreciated the attention to an important topic. Based on the reviews, we are likely to accept this manuscript for publication, providing that you modify the manuscript according to the review recommendations. 

Sincerely,

Jeremy Camp, Ph.D.

Deputy Editor

Jeremy Camp

Deputy Editor

Reviewer's Responses to Questions

**Key Review Criteria Required for Acceptance?**

**Methods**

-Are the objectives of the study clearly articulated with a clear testable hypothesis stated?

-Is the study design appropriate to address the stated objectives?

-Is the population clearly described and appropriate for the hypothesis being tested?

-Is the sample size sufficient to ensure adequate power to address the hypothesis being tested?

-Were correct statistical analysis used to support conclusions?

-Are there concerns about ethical or regulatory requirements being met?

Reviewer #1: The objectives of the study are clear, the study design is appropriate to address the stated objectives.

analyses used in the study were correct to support the conclusions.

Reviewer #2: (No Response)

**Results**

-Does the analysis presented match the analysis plan?

-Are the results clearly and completely presented?

-Are the figures (Tables, Images) of sufficient quality for clarity?

Reviewer #1: The results are clearly and completely presented.

Reviewer #2: (No Response)

**Conclusions**

-Are the conclusions supported by the data presented?

-Are the limitations of analysis clearly described?

-Do the authors discuss how these data can be helpful to advance our understanding of the topic under study?

-Is public health relevance addressed?

Reviewer #1: Conclusions supported the presented data. Authors also discuss how these data can be helpful to predicts the locations of WNF outbreaks better (e.g.spatio-environmental model) and the environmental model predicts the intensity of the WNV cases.

Reviewer #2: (No Response)

**Editorial and Data Presentation Modifications?**

Reviewer #1: Minor comments:

The influence of precipitation in the spring and early summer month would also be interesting to consider. The considered variables in the development of the models are annual precipitation and precipitation seasonality. Are they enough to get a good picture of the influence of precipitation? In addition, the influence of rain-fed agriculture on WNFV indicates, that this could be important to look into.

Reviewer #2: (No Response)

**Summary and General Comments**

Reviewer #1: Manuscript ID: PNTD-D-20-1456

Title: Predicting the spatio-temporal spread of the West Nile virus in Europe

The manuscript describes the development of risk models for spread of one of the most important emerging arboviral infections in Europe. The spatio-environmental model predicts the locations of WNF outbreaks best, and the environmental model predicts the intensity of the cases. The information gained in the manuscript is in-line with the previous knowledge. An English lecturer should read the introduction.

Minor comments:

The influence of precipitation in the spring and early summer month would also be interesting to consider. The considered variables in the development of the models are annual precipitation and precipitation seasonality. Are they enough to get a good picture of the influence of precipitation? In addition, the influence of rain-fed agriculture on WNFV indicates, that this could be important to look into.

Reviewer #2: The manuscript “Predicting the spatio-temporal spread of the West Nile virus in Europe ” is an original work aiming to provide insights on how environmental variables can be exploited to predict the spread of WNV in Europe. The topic is of interest. However, the length of the manuscript renders it quite difficult to focus and uncover the most important things. Additionally, several parts of the text are lengthy and irrelevant to the study – e.g. the first paragraph of the introduction, other phrases within that section, etc. The discussion is also rather extended (9 pages) and requires its re-reading several times in order to extract the important findings of this work. I suggest that the authors rearrange their manuscript by keeping only the important parts of it. 

Line 4. “West Nile virus” (and elsewhere in the manuscript). 

Line 5. The term “final hosts” applies for parasites, wherein a “final” host is required to complete the biological cycle. The proper term here is “dead-end” or “incidental”. 

Line 6. “Which affects the nervous system causing the West Nile Fever (WNF), leading to death”. a) The fact that the virus affects the nervous system (a condition called West Nile neuroinvasive disease, WNND) is not the same as fever. b) The way this has been written implies that WNV infection has always a lethal outcome. Please rephrase and use proper medical reports to present this sentence properly. This also applies for lines 31-32.

Lines 6-8. “Since the circulation of the virus in Europe in the 1950s, human cases have just increased, with 2018 having the highest number of cases registered to date”. The expression “human cases have just increased” is misleading, as “just” should refer to the last 25 years. The first large-scale epidemics in humans occurred in Romania during 1996, with 393 encephalitis cases in humans, including 16 deaths. Lineage 2 was first isolated in 2004 and increased numbers of cases due to this lineage were observed from 2010 onwards. This also applies for lines 32-33. 

Lines 44-54. This paragraph is irrelevant to the present study. The introduction should briefly summarize the current situation as well as what is lacking from literature regarding the studied topic. 

Line 55. “Flavivirus” and “Flaviviridae” should be italicized to comply with ICTV nomenclature rules. 

Lines 60-61. Reference to non-existent vaccines and drugs is not associated with the study. 

Lines 55-64. I suggest to reorganize and simplify this paragraph, by mentioning firstly how the virus is being transmitted, secondly the hosts, and lastly the condition in infected humans. 

Lines 65-77. This paragraph is also lengthy. The authors should help the reader focus to the main targets of their work and introduction should be written in a way to assist to this process. The fact, for example, that WNV took its name by the West Nile district of Uganda is distracting, not to mention that it is widely known. 

Line 119. “WNV-specific IgM”, etc. Use “serum (or virus) neutralization test” instead of “neutralization”. 

Line 275. “Rain-fed agriculture, as well as poultry and horse livestock were the human activities that also contributed to explain the distribution of the WNF” How do the authors interpret the finding regarding horses? Other factors are directly associated with the vector or the propagation of the virus itself (birds). So, could there be an explanation or hypothesis for this finding?

Line 517. “Horse density also appears to favor the outbreak of the disease. This may be caused by the fact that equids are also dead-end hosts of the virus cycle.” How can this be supported? In horses the virus acts as in humans (dead-end) and mosquitoes cannot get infected by an infected horse. So, the dead-end character of the infection of horses (and humans), which is an accidental thing, is not expected to favor the outbreaks.

PLOS authors have the option to publish the peer review history of their article (what does this mean?). If published, this will include your full peer review and any attached files.

Reviewer #1: No

Reviewer #2: No
---

## [Decision Letter · Decision Letter 1]

23 Nov 2020

Dear Mr. García-Carrasco,

Thank you very much for submitting your manuscript "Predicting the spatio-temporal spread of the West Nile virus in Europe" for consideration at PLOS Neglected Tropical Diseases. As with all papers reviewed by the journal, your manuscript was reviewed by members of the editorial board and by several independent reviewers. The reviewers appreciated the attention to an important topic. Based on the reviews, we are likely to accept this manuscript for publication, providing that you modify the manuscript according to the review recommendations. 

As mentioned previously, due to the Covid-19 pandemic, one first-round reviewer withdrew from the peer-review process. A third reviewer was invited and has provided a largely positive review with only a few minor suggestions for improvement. Thus, I would like to provide you with an opportunity to respond to the third reviewer's comments. 

Sincerely,

Jeremy Camp, Ph.D.

Deputy Editor

Reviewer's Responses to Questions

PLOS authors have the option to publish the peer review history of their article (what does this mean?). If published, this will include your full peer review and any attached files.

Reviewer #1: No

Reviewer #3: No

**Key Review Criteria Required for Acceptance?**

**Methods**

-Are the objectives of the study clearly articulated with a clear testable hypothesis stated?

-Is the study design appropriate to address the stated objectives?

-Is the population clearly described and appropriate for the hypothesis being tested?

-Is the sample size sufficient to ensure adequate power to address the hypothesis being tested?

-Were correct statistical analysis used to support conclusions?

-Are there concerns about ethical or regulatory requirements being met?

Reviewer #3: The objectives of the study are clearly elaborated and developed based on several established hypotheses. To address them, a well-calculated design has been made with several appropriate approaches. Although the methodology presented is extensive, it makes this analysis easier to understand. The population used for the predictive analysis is clearly described, supported by solid databases, with a sufficient sample size so that, in short, it is appropriate for the hypothesis analyzed in this work.

The statistical analysis used is correct to support the conclusions.

**Results**

-Does the analysis presented match the analysis plan?

-Are the results clearly and completely presented?

-Are the figures (Tables, Images) of sufficient quality for clarity?

Reviewer #3: The analysis presented is adjusted to the stated objectives, the results are clearly and completely presented, and the figures (tables, images) are of sufficient quality for clarity.

**Conclusions**

-Are the conclusions supported by the data presented?

-Are the limitations of analysis clearly described?

-Do the authors discuss how these data can be helpful to advance our understanding of the topic under study?

-Is public health relevance addressed?

Reviewer #3: The limitations of the analysis are clearly described throughout. The conclusions are supported by the data presented, with a subsequent discussion that, although extensive, is well developed, with a good relationship between results and available data on the incidence of WNV and its causes, so it is useful to advance in the prediction of future outbreaks, both of this virus and of other possible zoonotic ones. This work is, therefore, highly relevant for public health.

**Editorial and Data Presentation Modifications?**

Reviewer #3: Minor points:

- Delete the article in front of WNV and WNF throughout the entire document. Example: on line 4 replace "The West Nile virus ...." with "West Nile virus ....", on lines 46-47 replace "When infected by the WNV ...." with "When infected by WNV .... ", line 48 replace" that is known as the West Nile fever (WNF) "with" that is known as West Nile fever (WNF) ", or on line 312 replace" the WNF cases ..." by "WNF cases ... "

- line 8: better replace “in Europe” with “the continent” in order not to repeat “Europe” so many times.

- Replace throughout the document "the Julian week" by "the July week"

- line 33: “being 2018 the year with…”

- lines 60-62: (The transmission season of 2018 was exceptional; 1,605 cases were confirmed, which is twice as high as the sum of cases recorded in the previous three years): better “The transmission season of 2018 was exceptional; 1,605 cases were confirmed, which is double the sum of cases registered in the previous three years”

- line 72: “…are helpful for predicting the risk for WNF outbreaks” replace by “… are useful for predicting the risk of WNF outbreaks.”

- line 154: (“…added at each step if and only if the resulting new regression was…”) delete “if and”

- line 243: a comma is needed (“…the higher the spatial and environmental favorability, the earlier the cases would…”)

- lines 270, 285: delete “Error! Reference source not found.”

- line 424: change “Other studies also showed that the WNF incidence had a…” by “Other studies also showed that the incidence of WNF had a…”
---

## [Decision Letter · Decision Letter 2]

1 Dec 2020

Dear Mr. García-Carrasco,

We are pleased to inform you that your manuscript 'Predicting the spatio-temporal spread of West Nile virus in Europe' has been provisionally accepted for publication in PLOS Neglected Tropical Diseases.

Best regards,

Jeremy Camp, Ph.D.

Deputy Editor

Jeremy Camp

Deputy Editor

Reviewer's Responses to Questions

**Key Review Criteria Required for Acceptance?**

**Methods**

-Are the objectives of the study clearly articulated with a clear testable hypothesis stated?

-Is the study design appropriate to address the stated objectives?

-Is the population clearly described and appropriate for the hypothesis being tested?

-Is the sample size sufficient to ensure adequate power to address the hypothesis being tested?

-Were correct statistical analysis used to support conclusions?

-Are there concerns about ethical or regulatory requirements being met?

Reviewer #3: (No Response)

**Results**

-Does the analysis presented match the analysis plan?

-Are the results clearly and completely presented?

-Are the figures (Tables, Images) of sufficient quality for clarity?

Reviewer #3: (No Response)

**Conclusions**

-Are the conclusions supported by the data presented?

-Are the limitations of analysis clearly described?

-Do the authors discuss how these data can be helpful to advance our understanding of the topic under study?

-Is public health relevance addressed?

Reviewer #3: (No Response)

**Editorial and Data Presentation Modifications?**

Reviewer #3: (No Response)

**Summary and General Comments**

Reviewer #3: (No Response)

PLOS authors have the option to publish the peer review history of their article (what does this mean?). If published, this will include your full peer review and any attached files.

Reviewer #3: No

---

## [Editor Report · Acceptance letter]

21 Dec 2020

Dear Mr. García-Carrasco,

We are delighted to inform you that your manuscript, "Predicting the spatio-temporal spread of West Nile virus in Europe," has been formally accepted for publication in PLOS Neglected Tropical Diseases.

Best regards,

Shaden Kamhawi

co-Editor-in-Chief

Paul Brindley

co-Editor-in-Chief
